# Impact of Microbiota Depletion by Antibiotics on SARS-CoV-2 Infection of K18-hACE2 Mice

**DOI:** 10.3390/cells11162572

**Published:** 2022-08-18

**Authors:** Patrícia Brito Rodrigues, Giovanni Freitas Gomes, Monara K. S. C. Angelim, Gabriela F. Souza, Stefanie Primon Muraro, Daniel A. Toledo-Teixeira, Bruna Amanda Cruz Rattis, Amanda Stephane Passos, Laís Passarielo Pral, Vinícius de Rezende Rodovalho, Arilson Bernardo dos Santos P. Gomes, Valquíria Aparecida Matheus, André Saraiva Leão Marcelo Antunes, Fernanda Crunfli, Krist Helen Antunes, Ana Paula Duarte de Souza, Sílvio Roberto Consonni, Luiz Osório Leiria, José Carlos Alves-Filho, Thiago M. Cunha, Pedro M. M. Moraes-Vieira, José Luiz Proença-Módena, Marco Aurélio R. Vinolo

**Affiliations:** 1Laboratory of Immunoinflammation, Institute of Biology, University of Campinas (UNICAMP), Campinas 13000-000, Brazil; 2Center of Research in Inflammatory Diseases, Ribeirão Preto Medical School, University of São Paulo, Ribeirão Preto 14000-000, Brazil; 3Laboratory of Immunometabolism, Institute of Biology, University of Campinas (UNICAMP), Campinas 13000-000, Brazil; 4Laboratory of Emerging Viruses, Institute of Biology, University of Campinas (UNICAMP), Campinas 13000-000, Brazil or; 5Department of Pathology, Faculty of Medicine of Ribeirão Preto, University of São Paulo, Ribeirão Preto 14000-000, Brazil; 6Center for Research in Inflammatory Diseases (CRID), Department of Pharmacology, Ribeirão Preto Medical School, University of São Paulo, Ribeirão Preto 14000-000, Brazil; 7Laboratory of Neuroproteomics, Institute of Biology, University of Campinas (UNICAMP), Campinas 13000-000, Brazil; 8Laboratory of Clinical and Experimental Immunology, Pontifical Catholic University of Rio Grande do Sul, Porto Alegre 90000-000, Brazil; 9Laboratory of Citochemistry and Immunocitochemistry, Institute of Biology, University of Campinas (UNICAMP), Campinas 13000-000, Brazil; 10Obesity and Comorbidities Research Center (OCRC), University of Campinas (UNICAMP), Campinas 13000-000, Brazil; 11Experimental Medicine Research Cluster, University of Campinas (UNICAMP), Campinas 13000-000, Brazil

**Keywords:** respiratory infection, SARS-CoV-2, COVID-19, intestinal microbiota, colon, gut-to-lung axis

## Abstract

Clinical and experimental data indicate that severe acute respiratory syndrome coronavirus (SARS-CoV)-2 infection is associated with significant changes in the composition and function of intestinal microbiota. However, the relevance of these effects for SARS-CoV-2 pathophysiology is unknown. In this study, we analyzed the impact of microbiota depletion after antibiotic treatment on the clinical and immunological responses of K18-hACE2 mice to SARS-CoV-2 infection. Mice were treated with a combination of antibiotics (kanamycin, gentamicin, metronidazole, vancomycin, and colistin, Abx) for 3 days, and 24 h later, they were infected with SARS-CoV-2 B lineage. Here, we show that more than 80% of mice succumbed to infection by day 11 post-infection. Treatment with Abx had no impact on mortality. However, Abx-treated mice presented better clinical symptoms, with similar weight loss between infected–treated and non-treated groups. We observed no differences in lung and colon histopathological scores or lung, colon, heart, brain and kidney viral load between groups on day 5 of infection. Despite some minor differences in the expression of antiviral and inflammatory markers in the lungs and colon, no robust change was observed in Abx-treated mice. Together, these findings indicate that microbiota depletion has no impact on SARS-CoV-2 infection in mice.

## 1. Introduction

Severe acute respiratory syndrome coronavirus-2 (SARS-CoV-2) is the etiological agent of coronavirus disease-2019 (COVID-19), a disorder that affects the respiratory tract, triggering a severe respiratory disorder and pneumonia in humans [1]. This virus emerged in Wuhan City, China, at the end of 2019 and quickly spread to all continents. By March 2022, more than 458 million cases had been confirmed, with over 6 million deaths due to COVID-19, according to the Worldometer Coronavirus database [2].

Clinical manifestations of COVID-19 vary from asymptomatic cases to severe disease [3]. Comorbidities such as obesity, diabetes, hypertension and cardiovascular diseases, older age and immunocompromised states have been strongly linked with severe outcomes [4]. Multiple vaccines against SARS-CoV-2 have been successfully developed and offered to the population [5], having a significant positive impact on the number of cases and deaths [6].

SARS-CoV-2 pathogenesis involves different steps: (1) virus entry and replication in in epithelial, endothelial and immune cells of the respiratory tract, (2) destruction of infected cells with virus release and (3) triggering of a local immune response, which may be sufficient to eliminate the infection [7]. In some cases, the infection may evolve to a dysfunctional immune response that leads to lung damage, endothelial dysfunction and systemic alterations, resulting in severe clinical complications such as abnormal blood coagulation, heart diseases, neurological alterations and liver and kidney injuries, which can progress to multi organ failure and death [7]. Infection by SARS-CoV-2 is not limited to the lungs and respiratory-associated tissues. Recent studies have demonstrated that gastrointestinal (GI) manifestations including loss of appetite, nausea or vomiting, diarrhea and abdominal pain are relatively common in SARS-CoV-2-infected patients [8]. It has also been reported that the GI tract appears to be an important target of SARS-CoV-2 replication, since viral mRNA and the SARS-CoV-2 nucleocapsid protein have been frequently detected in different parts of the human GI tract in infected individuals [9,10]. Moreover, human intestinal epithelial cell lines [11] are usually susceptible to SARS-CoV-2 and this virus also infects and replicates in human small intestine enterocytes [12]. GI manifestations of SARS-CoV-2 infection can be the result of direct histopathological alterations, but may also reflect the systemic effects of the infection or changes induced in the immune system or the intestinal microbiota [13,14].

The GI tract is the largest immunological tissue in the body and its resident microbiota modulate host immune responses [15]. Promising results obtained using different models demonstrated the relevance of the intense and complex cross-talk between the gut microbiota, the lungs and the systemic immune response, thus highlighting the potential of preventive and/or therapeutic strategies for respiratory infectious diseases based on changes in microbiota composition or the production of metabolites [16,17,18,19,20]. 

COVID-19 patients present drastic changes in gut microbiota composition, including an increased amount of opportunistic pathogens such as *Clostridium hathewayi*, *Actinomyces viscosus* and *Bacteroides nordii* [21]. Depletion of beneficial commensals including *Faecalibacterium prausnitzii*, *Lachnospiraceae bacterium* 5_1_63FAA, *Eubacterium rectale*, *Ruminococcus obeum* and *Dorea formicigenerans* has been observed in COVID-19 patients treated with antibiotics, indicating that this intervention can accentuate the shift in microbiota composition from a healthy to an unhealthy condition [21]. The dysbiotic gut microbiota of COVID-19 patients has been associated with elevated levels of cytokines and inflammatory markers, suggesting a relationship between alterations in gut microbiota and the severity of the disease [22]. Moreover, a recent study found an association between COVID-19 gut dysbiosis, particularly depletion of *Faecalibacterium* and *Roseburia* genera, and an increased inflammatory profile, as observed in severe or critical COVID-19 patients [23]. A recent study found an association between dysbiotic microbiota and the translocation of bacteria into the blood of COVID-19 patients, thus contributing to the increased inflammatory profile observed in these patients and the development of secondary infections [24]. Intestinal dysbiosis has also been reported in mice, hamsters and nonhuman primates infected with SARS-CoV-2 [25,26,27], indicating that the gut microbiome profile is involved in this disease and strategies to alter the intestinal microbiota might change the disease outcome.

The relationship between COVID-19 and changes in the composition of the intestinal microbiota becomes increasingly evident with the advancement of research in humans and animal models. However, factors associated with treatment during COVID-19 in humans make it difficult to understand the role of the microbiota in the development of the disease. In the present study, we explored the effect of acute treatment with broad-range oral antibiotics, which was previously demonstrated to impair the antiviral response in mice to respiratory syncytial virus (RSV) [28]. We show that treatment with antibiotics has no direct impact on the survival and immune response of SARS-CoV-2-infected mice. Moreover, microbiota depletion had no significant effects on viral lethality, tropism, load and histopathological alterations in key target tissues.

## 2. Materials and Methods

### 2.1. Animals

Adult heterozygous K18-hACE2 transgenic female mice were purchased from the Multidisciplinary Center for Biological Investigation (CEMIB), Campinas, São Paulo, Brazil. Mice were kept in regular filter-top cages with free access to sterile water and food. Animal procedures were approved by the Ethics Committee on Animal Use of the University of Campinas (protocol #5495-1/2020). 

### 2.2. Antibiotic Treatment

Mice were provided with sterile drinking water supplemented with an antibiotic mix (Abx) for three days before SARS-CoV-2 infection. Abx [28] was composed of kanamycin (0.4 mg/mL), gentamicin (0.035 mg/mL), metronidazole (0.045 mg/mL), vancomycin (0.045 mg/mL) and colistin (0.035 mg/mL), purchased from Sigma-Aldrich (St. Louis, MO, USA). The addition of antibiotics to drinking water did not cause a reduction in water intake by the animals and no diarrhea was observed.

### 2.3. Virus

SARS-CoV-2 B strain (HIAE-02-SARS CoV-2/SP02/human/2020/BRA; GenBank MT126808.1) was a gift from Prof. Dr. Edison Durigon (ICB-USP, São Paulo, Brazil). Viral stock was propagated in Vero cells (ATCC CCL81) and the supernatant was harvested at 2–3 days post-infection (dpi) and kept at −80 °C. Viral titers were determined by plaque assays and RT-qPCR [29]. The virus was produced in a biosafety level (BSL) 3 area of the Laboratory of Emerging Viruses and kindly provided by Prof. Dr. José Luiz Proença-Módena (Institute of Biology, UNICAMP, Campinas, Brazil).

### 2.4. SARS-CoV-2 Infection and Clinical Analysis of Mice

Virus inoculation was performed under anesthesia induced using a mixture of xylazine and ketamine (20 and 100 mg/kg, respectively). Mice were intranasally administered 5 × 10^4^ plaque forming units (PFU) of SARS-CoV-2 in a total volume of 40 μL for mortality testing and 1 × 10^4^ PFU for the other experiments involving tissue harvesting at 5 dpi. All infections were performed at the animal BSL 3 laboratory of Ribeirão Preto Medical School (FMRP, Ribeirão Preto, Brazil). Following infection, mice were monitored daily for body weight changes and signs of disease. A clinical score based on body weight variation, behavior and respiratory distress was used to evaluate the disease over the course of infection in each animal (Table 1). The clinical score was obtained by summing the scores of the six parameters evaluated for each animal. The clinical score data are presented as the mean of the group per day.

### 2.5. Bacterial DNA Isolation from Mice Feces

Fecal samples were collected 1day before and 1day after (D0) antibiotic treatment and prior to SARS-CoV-2 infection. Fecal pellets were collected in sterile DNAse-free tubes and immediately frozen in liquid nitrogen. Samples were kept at −80 °C until use. DNA extraction was performed using the Purelink Microbiome DNA purification kit (ThermoFisher Scientific, Waltham, MA, USA), following the manufacturer’s recommendations. Purified DNA was eluted in 20 μL elution buffer and kept at −20 °C. DNA concentrations were measured using a NanoDrop 2000 spectrophotometer.

### 2.6. rRNA Sequencing and Analysis

Luminal colonic contents were collected at 5 dpi from both experimental groups. In addition, we also included samples from non-infected animals. Samples were collected in sterile DNAse-free tubes and immediately frozen in liquid nitrogen. DNA extraction was performed using the Purelink Microbiome DNA purification kit (ThermoFisher Scientific), following the manufacturer’s recommendations. Purified DNA was eluted in 20 μL elution buffer and kept at −20 °C. DNA concentrations were measured using a NanoDrop 2000 spectrophotometer. Universal primers 341F (5′-CCT AYG GGR BGC ASC AG-3′) and 806R (5′-GGA CTA CNN GGG TAT CTA AT-3′) were used for the amplification of the V3-V4 region of the bacterial 16S rRNA gene. Library quantification and quality was assessed on Qubit@ 2.0 Fluorometer (Thermo Scientific) and Agilent Bioanalyzer 2100 systems. The libraries were sequenced on a NovaSeq PE250 at Novogene.

Fastq files with raw sequences were subjected to quality control with FastQC [32] and MultiQC [33]. Demultiplexed sequences were imported into QIIME2 2021.11 [34], for filtering, paired-end read combination, denoising and chimera detection with the DADA2 plugin [35]. The resulting table of amplicon sequence variants (ASVs) was rarefied at a rarefaction depth of 42,258 [36] and used to construct a phylogenetic placement with SEPP [37]. Alpha and beta diversity metrics, as well as Principal Coordinate Analysis (PCoA), were estimated with the QIIME2 diversity plugin [38,39,40,41]. Taxonomic composition analysis was performed with the q2-feature-classifier [42] using a Naive Bayes classifier trained on Silva 138 99% OTU full-length sequences [43,44,45]. Differential analysis was conducted at phylum and class level using the QIIME2 composition plugin with the ANCOM statistical framework [46]. Visualizations were obtained with the QIIME2 View interface and Python’s library Seaborn [47]. The raw reads of 16S rRNA sequencing were submitted to the National Center for Biotechnology Information’s Sequence Read Archive (NCBI SRA) with the accession number PRJNA858922.

### 2.7. RNA Extraction and Quantification

Tissues were weighed and homogenized in 1 mL Hank’s Balanced Salt (HBSS) containing a mixture of antibiotics (0.2% Normocin, 1% Penicillin and Streptomycin and 1% Gentamicin) and zirconia beads in MagNaLyser equipment (Roche Life Science, Mannheim, Germany). The homogenates were clarified by centrifugation and used for RNA extraction, which was performed using the Quick-RNA viral kit (Zymo Research, Irvine, CA, USA), following the manufacturer’s instructions. RNA concentrations were determined using the NanoDrop 2000 spectrophotometer. 

### 2.8. Viral Load Quantification

Viral RNA was detected and quantified by one-step RT-qPCR, using primers for gene E (envelope protein), as previously described [48]. Briefly, all assays were performed using TaqMan Fast Virus 1-Step Master Mix (Applied Biosystems, Waltham, MA, USA), 800 nM of primers (F: 5-ACA GGT ACG TTA ATA GTT AAT AGC GT-3; R: 5-ATA TTG CAG CAG TAC GCA TAC GCA CAC A-3), 400 nM of probe (P: 5-6FAM-ACA CTA GCC ATC CTT ACT GCG CTT CG-QSY-3) and 6 μL RNA samples. The PCR cycling conditions were: 1 cycle of 50 °C for 10 min, 1 cycle of 95 °C for 2 min, followed by 45 cycles of 95 °C for 5 s and 60 °C for 30 s, using the QuantStudio3 Real-Time PCR System (Applied Biosystems). Negative samples and a standard curve were included in all PCR runs. The standard curve was built using serial 10-fold dilutions of viral stock of known titer, and the viral copy number was expressed on a log10 scale as viral RNA equivalents per gram or per milliliter after normalization for tissue weight.

### 2.9. Histopathological Score

After 5 dpi, mice were anesthetized with ketamine–xylazine mixture (20 and 100 mg/kg, respectively) and, after total loss of response to stimuli, blood was collected through the retrobulbar venous plexus. Mice were intracardially perfused with 20 mL sterile saline solution before harvesting the tissues of interest. For histological analyses, samples were fixed in 4% PFA for 72 to 96 h at 8 °C. After fixation, tissues were washed three times in saline solution and kept in 70% ethanol solution. Lungs were embedded in paraffin and sectioned transversely at a width of 5 µm. Intestines were embedded in historesin (Leica Microsystems, Heidelberg, Germany) and sectioned transversely at a width of 2 µm. Sections were produced using a microtome for hematoxylin and eosin staining. On intestine sections, the presence of inflammatory cell infiltration, edema and epithelial erosions was assessed. For lung tissue analysis, factors such as type II pneumocyte hyperplasia, perivascular, septal and alveolar inflammation, edema and alveolar fibrin were evaluated [49].

### 2.10. Bronchoalveolar Lavage Fluid (BALF)

Mice were anesthetized as described below and the tracheas were cannulated. The lungs were washed with a cold DMEM medium. BALF were kept on ice until all samples were collected. Samples were centrifuged and pellets suspended for total cell count and flow cytometry analysis. The counting procedure was performed in a blinded manner by an experienced investigator. 

### 2.11. Flow Cytometry

After washing with saline solution, single cell suspensions were stained for 20 min at 4 °C, with two separate staining mixtures: (a) anti-CD45-APC-Cy7 (#103116 BioLegend^®^ (San Diego, CA, USA), Clone 30-F11), anti-CD3ε-PerCP-Cy5.5 (#100217 BioLegend^®^, Clone 17A2), anti-CD4-PE (#553048 BD Biosciences^®^ (Franklin Lakes, NJ, USA), Clone RM4-5), anti-CD8a-APC (#100711BioLegend^®^, Clone 53-6.7) or (b) anti-CD45-APC-Cy7 (#103116 BioLegend^®^, Clone 30-F11), anti-CD11c-APC (#117309BioLegend, clone N418), anti-CD11b-PE (#101207BioLegend^®^, Clone M1/70), anti-LyG-FITC (#127605BioLegend^®^, Clone 1A8) and anti-NK1.1-Brilliant Violet 605 (#108739, BioLegend^®^, Clone PK136). After washing in FACS buffer (saline solution with 1% fetal bovine serum), cells were fixed in 4% paraformaldehyde for 30 min at room temperature before taking samples out of the BSL3 area. Cells were again washed, resuspended in FACS buffer and acquired using FACS Symphony A5 with BD FACS Diva software. Cell populations were gated and quantified by FlowJo10.7 software. Gating strategies used for analysis are presented in Appendix A.

### 2.12. Enzyme-Linked Immunosorbent Assay (ELISA)

Lung and colon fragments (15–30 mg) were collected in 300 μL RIPA buffer supplemented with protease inhibitors. Samples were processed for protein extraction on ice using a tissue homogenizer and then centrifuged at 10,000 rpm/10 min/4 °C. The supernatant was collected for enzyme-linked immunosorbent assay (ELISA). The levels of TNF-α, IL-6, IL-1β, CXCL1, CXCL2, IFN-β (lung and colon) and IL-17 (colon) were measured following the manufacturer’s recommendations (R&D Systems, Minneapolis, MN, USA). The results were expressed in pg/mL and normalized to the total protein content, as determined by Bradford assay (Bio-Rad, Hercules, CA, USA). Concentrations of lipocalin-2 in the luminal content of the colon were determined with an ELISA kit, according to the manufacturer’s instructions (R&D Systems). The concentrations of this protein were expressed in pg/mg of luminal content.

### 2.13. Quantitative Real-Time PCR Analysis

RNA was reverse-transcribed using the High-Capacity RNA-to-cDNA™ Kit (Thermo Fisher) according to the manufacturer’s instructions. qRT-PCR was performed using the Sybr Green Master mix (Applied Biosystems™, Walthan, MA, USA) and the BIO-RAD CFX394 Touch Real-Time PCR Detection System. Relative gene expression was calculated using the ΔΔCt method with the 18S gene as a reference. The sequences of the primers used are given in Table 2.

### 2.14. Statistical Analysis

Statistical analyses were performed using GraphPad Prism 8.0 software (San Diego, CA, USA). Results are presented as mean ± standard error mean (SEM). For comparison between 2 groups, Student’s *t*-test was used. For comparison between more than 2 groups, one-way ANOVA followed by Tukey’s post hoc test analysis was applied. For analysis with more than 2 variables, two-way ANOVA was applied. Differences were considered statistically significant for *p* values < 0.05.

## 3. Results

### 3.1. Microbiota Depletion Does Not Change Mortality of SARS-CoV-2 Infection, but Alters Clinical Symptoms

To explore the role of the intestinal microbiota in SARS-CoV-2 infection, we conducted a series of experiments in mice previously treated with an oral antibiotic cocktail (Abx), a treatment known to deplete most of the bacteria present in the guts of mice [30]. Briefly, female K18-hACE2 mice received the antibiotics in drinking water for 3 days before SARS-CoV-2 intranasal inoculation (Figure 1A). The efficacy of the Abx treatment was confirmed by macroscopic analysis of the cecum and colon, which were enlarged, and by measurement of bacterial DNA load in the feces of mice on the day of infection. Fecal DNA was nearly five-fold lower in Abx-treated mice (Appendix A). Despite these effects, we did not observe diarrhea or any clinical alteration in mice under Abx treatment. 

Next, we analyzed the impact of the Abx treatment on infection-associated lethality. We infected Abx and control mice with 5 × 10^4^ PFU/animal of SARS-CoV-2 and followed these infected mice until 12 dpi (Figure 1A). Abx treatment had no significant effect on mouse mortality compared with the control group (Figure 1B). We repeated the infections using lower titers of virus (1 × 10^4^ PFU/animal) and followed the development of clinical signs until 5 dpi. Despite the fact that mice in the Abx and control groups presented a similar body weight loss during the course of infection (Figure 1C), a more intense deterioration in clinical signs was observed in controls (Figure 1D). The effect on Abx-treated mice was not associated with a significant difference in viral load in the tissues analyzed or the colon luminal content (Figure 1E). The microbiota composition of Abx-treated infected mice at 5 dpi presented significant differences in abundance and evenness (i.e., reduction in Abx-treated mice compared to the other groups) (Appendix A). In weighted UniFrac-based beta diversity analysis, we found a significant difference between non-infected and infected groups, but not between Abx and control infected mice (Appendix A). The bacterial relative abundance was different between experimental groups: compared to the non-infected group, we found a higher proportion of the *Verrucomicrobiota* phylum in the infected groups (Appendix A), which was associated with an increase in the *Akkermansia* genus. This alteration was previously reported in SARS-CoV-2-infected mice [27]. Reductions in *Firmicutes* (mainly due to a lower proportion of *Bacilli* class) were also observed when comparing infected groups with the non-infected group. Moreover, the proportion of the *Desulfobacterota* phylum (and *Desulfovibrionia* class) was increased in the control group (compared to the non-infected group) and decreased in the Abx group (relative to the control group). Increases in the *Desulfobacterota* phylum were also verified in the fecal microbiota of SARS-CoV-2-infected hamsters [26]. Together, these results indicate that SARS-CoV-2 infection results in significant changes in the bacterial communities of the gut and that Abx treatment has only minor effects on the development of SARS-CoV-2 infection in mice.

### 3.2. Antibiotic Treatment Does Not Affect Lung Histopathology but Promotes Changes in Production of Immune Molecules Following SARS-CoV-2 Infection

We next evaluated the impact of Abx on lung histopathology and inflammatory response during SARS-CoV-2 infection. At 5 dpi, we observed intense histopathological alterations on hematoxylin and eosin-stained lung sections of infected mice, including the accumulation of immune cells in different locations, mainly in perivascular areas, and alveolar wall thickening (Figure 2A). No difference was observed in histopathological parameters between infected control and Abx mice (Figure 2B). 

To characterize the lung inflammatory profile, we assessed the immune cells by flow cytometry of bronchoalveolar lavage fluid (BALF) and measured the mRNA and protein levels of pro-inflammatory and anti-viral cytokines in lung tissue. The gut microbiota reduction by Abx significantly reduced the total cell number in the BALF, mainly attributed to both lymphocytes CD4+ and CD8+ (Figure 2C). No significant effect of Abx was found on innate immune cells in the BALF. Among the cytokines analyzed, the only difference observed between the experimental groups was a reduction in the levels of IL-1β in the lung homogenates of Abx-treated mice (Figure 2D). We also found increased expression of *Ifna* in the lungs of Abx-treated mice compared to controls (Figure 2E). No significant changes in the levels or expression of other inflammatory cytokines, chemokines or antiviral molecules were observed between the experimental groups (Figure 2D,E). 

### 3.3. Antibiotic Treatment Increases Inflammatory Cytokines in Colon but This Effect Is Not Associated with Significant Alterations in Colon Histopathology

A previous study showed that human ACE2 is highly expressed along the intestinal tract (stomach to large intestine) of K18-hACE2 mice [50]. This explains the presence of viral RNA at relatively high levels in these tissues [50]. However, we observed minor histopathological changes in hematoxylin and eosin-stained colon sections of infected mice (at 5 dpi), including an increment in the number of nuclei at the crypt base, pointing to an increase in proliferation, and a reduction in mucus production. In the histological sections of Abx-treated mice, we only observed a reduction in mucus compared with non-infected mice. No infiltration of inflammatory cells or exfoliation of epithelial cells was observed in infected mice (Figure 3A).

We next examined the levels of pro-inflammatory cytokines in the colon, along with measurement of lipocalin-2 in the luminal content of colon samples. In accordance with the histopathological score, there was no significant difference in the lipocalin-2 levels at 5 dpi between the infected groups or between these and non-infected mice (Figure 3B). Nevertheless, the colons of Abx-treated mice showed an increase in CXCL-2 and IL-17 cytokines, compared with the control group (Figure 3C). The other cytokines analyzed presented similar concentrations among infected animals, treated or not with the antibiotic cocktail prior to infection.

## 4. Discussion

Recent studies indicate that a large proportion (more than threequarters) of COVID-19 patients are treated with antibiotics [51], [52]. This is much higher than the prevalence of bacterial co-infections [51], thus indicating the unnecessary use of this therapy, which can accentuate the gut dysbiosis induced by SARS-CoV-2 infection and increase the susceptibility to *Clostridioides difficile* infections. A number of studies have observed an exacerbated disease in antibiotic-treated mice infected with respiratory pathogens including *Pseudomonas aeruginosa* [53], influenza virus [54] and respiratory syncytial virus (RSV) [28]. Antibiotic treatment has been shown to impair antiviral responses, thus rendering mice more susceptible to infection by multiple viruses, including West Nile (WNV), Dengue and Zika virus infection [55].

We found that acute microbiota depletion by oral antibiotics had no impact on SARS-CoV-2 mortality in mice. In agreement with these findings, we did not observe changes in viral load or histopathological alterations in the lungs or colons of infected mice. The experimental model used in our study, transgenic mice expressing the human ACE2 receptor under control of cytokeratin-18 promoter (K18-hACE2), develops a severe viral disease after SARS-CoV-2 inoculation. Mice infected with 5 × 10^4^ CFUs began to succumb at 7–8 dpi and only 20% survived until the 12th dpi. These results are in agreement with previous data obtained by other groups using this experimental model [49,50,56]. Despite no significant impact on survival, we observed that Abx-treated mice had a less intense deterioration in clinical signs compared to control mice.

Abx-treated mice presented a reduction in IL-1β concentrations and an increase in *Ifna* expression in the lungs. Both cytokines are relevant for SARS-CoV-2 response and pathogenesis. IL-1β is one of the proinflammatory cytokines excessively produced in the acute phase of SARS-CoV-2 infection. A recent experimental study highlighted the relevance of this cytokine for lung damage and indicated that blocking it may have a protective effect [57]. In contrast with the harmful effects of IL-1B, intranasal administration of IFN-A had a beneficial effect on SARS-CoV-2-infected Syrian hamsters [58]. In addition, we found a reduction in T cells present in the BAL of Abx-treated mice. Contrary to our initial hypothesis, these results suggest that microbiota depletion may have attenuating effects towards SARS-CoV-2 infection. This may be due to a reduction in bacteria translocation after antibiotic treatment. A recent study reported that COVID-19 patients present higher levels of markers associated with gut leakage than healthy individuals, and the levels of these markers are higher in patients with a more severe form of COVID-19 compared with patients with less severe forms of this infection [59]. In addition, the dysbiotic microbiota of COVID-19 patients has been associated with the development of secondary bloodstream infections [24]. It is worth mentioning that Abx treatment reduces the gut bacterial load, thus diminishing the amount of microorganisms that interact with the epithelium and may cross this barrier. However, it may also select some microorganisms, thus contributing to the development of secondary resistant infections, as has been reported for COVID-19 patients [60].

We also found an increase in IL-17 and CXCL-1 levels in the colons of Abx-treated mice. However, no change in inflammatory cells was observed between the groups at the time point analyzed. A similar increment in pro-inflammatory cytokines (e.g., CXCL1, CXCL2 and IL-1β) was previously reported in *E. histolytica*-infected mice treated with Abx, compared to mice that were infected but not treated with Abx [61]. The authors of the mentioned study demonstrated that these alterations were not associated with an increment in neutrophil migration to the intestine because the Abx treatment reduced the responsiveness of neutrophils to chemokines [61], an effect that may also be present in our model. 

The dysbiosis associated with the antibiotic regimen used in this study has been well characterized in previous studies [28,62,63]. In addition to drastic changes in the gut microbiota profile, including a reduction in microbial diversity and significant changes in community structure, this treatment was also associated with a reduction in the production of several microbiota-derived metabolites relevant for the host immune responses, including short-chain fatty acids (SCFAs). Moreover, COVID-19 patients who received antibiotics showed more intense alterations in gut microbiota composition compared to those who were not exposed to antibiotics, with a decrease in multiple beneficial symbionts, including the SCFA-producers *Faecalibacterium prausnitzii*, *Lachnospiraceae bacterium* 5_1_63FAA, *Eubacterium rectale*, *Ruminococcus obeum* and *Dorea formicigenerans* [21]. SCFAs are an important link between microbiota and immunity. As already mentioned, COVID-19 dysbiotic microbiota produce lower amounts of these metabolites, which have been linked with positive effects in different infection models [28,53,54,62,64]. However, recent evidence indicates that the SCFAs have no impact on SARS-CoV-2 infection [17,26,65] and may also not be effective in other viral infections [66], indicating that their effect is limited to specific infectious agents. Our data on Abx-treated mice indirectly support the hypothesis that SCFAs play no relevant role in SARS-CoV-2 infection.

Several aspects remain to be investigated in the context of the gut–lung axis during SARS-CoV-2 infection. For example, is SARS-CoV-2 intestinal dysbiosis secondary to the systemic inflammation or is it a direct effect of the virus on intestinal epithelial cells, thus affecting the interaction between them (and immune cells) and the components of the microbiota? Two other important questions that need to be addressed in this context are (1) whether the dysbiotic microbiota associated with diabetic and obese individuals play a role in this process and (2) whether dietary interventions that are known to affect microbiota composition and function may have an impact on disease progression. Many dietary strategies that allow us to verify the role of the microbiota in the regulation of the immune system during respiratory diseases such as probiotic therapy [67,68,69], intestinal microbiota transplantation [70], the use of fermentable dietary fibers (prebiotics) [70,71,72] or through supplementation with compounds of bacterial metabolism (e.g., SCFAs) [28,70,71] can be evaluated in future studies on SARS-CoV-2. Our study has limitations, including the absence of analyses on the effect of Abx treatment during the course of infection, or treatment with more potent Abx regimens or antibiotics that have been commonly used in COVID-19 patients, such as azithromycin. In addition, we only focused our analysis on a limited number of time points and doses and we did not test SARS-CoV-2 variants.

## 5. Conclusions

In summary, our data demonstrate that gut microbiota depletion by acute treatment with a broad range of oral antibiotics did not alter survival and had only minor effects on the immune response to SARS-CoV-2 in K18-ACE2 mice.

## Figures and Tables

**Figure 1 cells-11-02572-f001:**
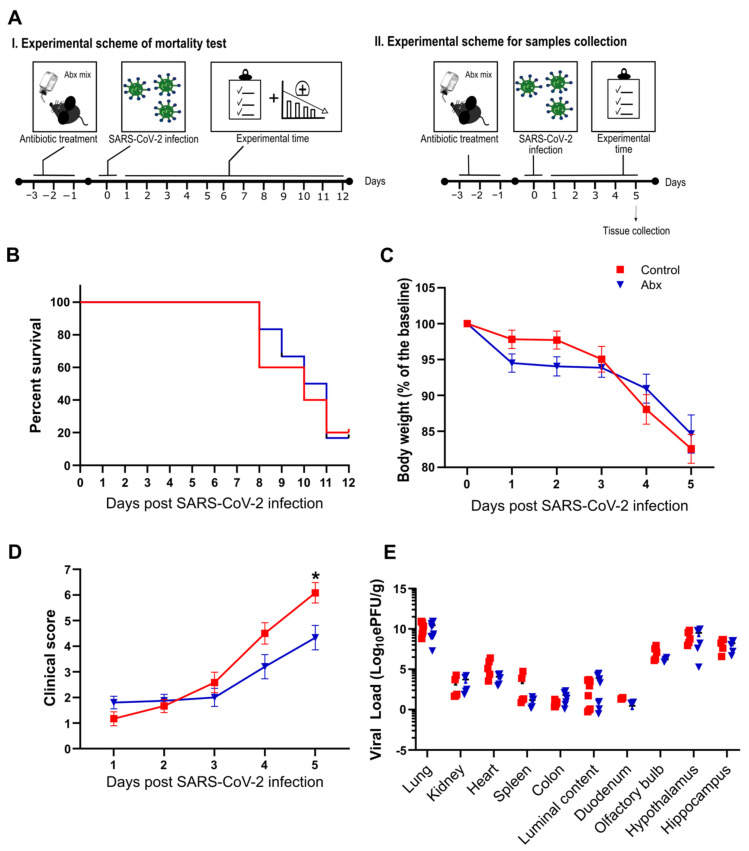
(**A**) Female K18-hACE2 mice were either treated (Abx) or not treated (control) for 3 days before SARS-CoV-2 infection. Body weight and clinical scores were measured after infection. I. Experimental scheme of mortality: experiments were performed to evaluate percent survival up to 12 dpi. II. Experimental scheme for sample collection: mice were euthanized at 5 dpi and the organs were collected for viral load quantification. (**B**) Survival rate after infection with SARS-CoV-2 (*n* = 6). (**C**) Body weight changes after infection (*n* = 12–15). (**D**) Clinical scores of infected mice (*n* = 12–15). * *p* < 0.05, 2-way ANOVA. (**E**) Viral load determination by RT-qPCR. * *p* < 0.05 by Student’s *t*-test (*n* = 4–10).

**Figure 2 cells-11-02572-f002:**
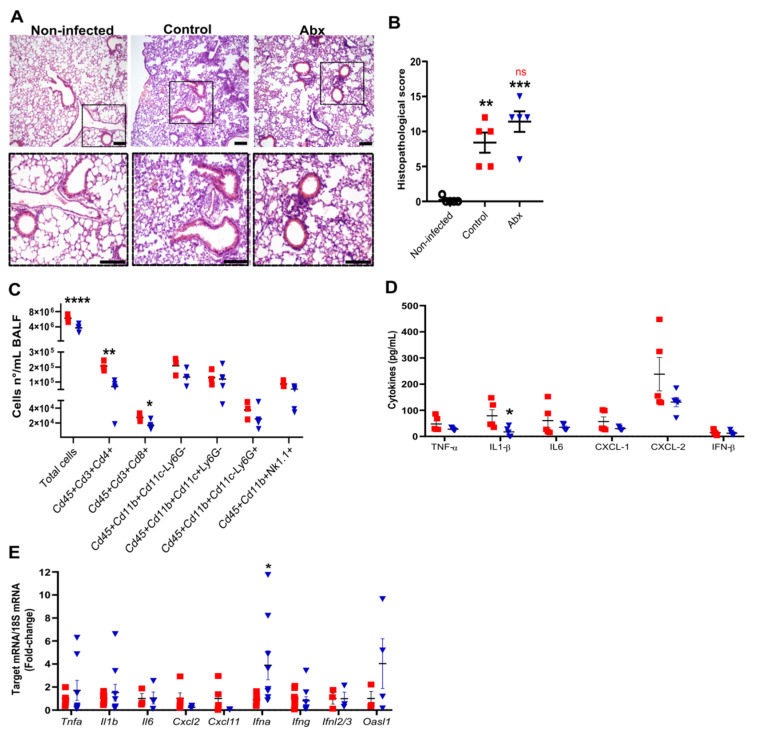
Lung histological and inflammatory alterations after infection. (**A**) Representative images of hematoxylin and eosin staining of lung sections from each experimental group. Scale bar = 100 µm. (**B**) Lung histopathological scores from each experimental group. ** *p* < 0.005 and *** *p* < 0.001 by 2-way ANOVA; Sidak’s multiple comparisons test (*n* = 5). (**C**) Total cell number and differential cell counts of lymphocytes (Cd45+Cd3+Cd4+ and Cd45+Cd3+Cd8+), myeloid (monocytes Cd45+Cd11b+Cd11c−Ly6G−, dendritic cells Cd45+Cd11b+Cd11c+Ly6G− and neutrophils Cd45+Cd11b+Cd11c−Ly6G+) and NK cells (Cd45+Cd11b+NK1.1+) in BALF. **** *p* < 0.0001, ** *p* < 0.005 and * *p* < 0.05, Student’s *t*-test (*n* = 3–5). (**D**) Concentration of cytokines in lung homogenates, as measured by ELISA. * *p* < 0.05, Student’s *t*-test (*n* = 5). (**E**) Expression of inflammatory and antiviral genes quantified using RT-qPCR in lung samples. * *p* < 0.05, Student’s *t*-test (*n* = 3–9).

**Figure 3 cells-11-02572-f003:**
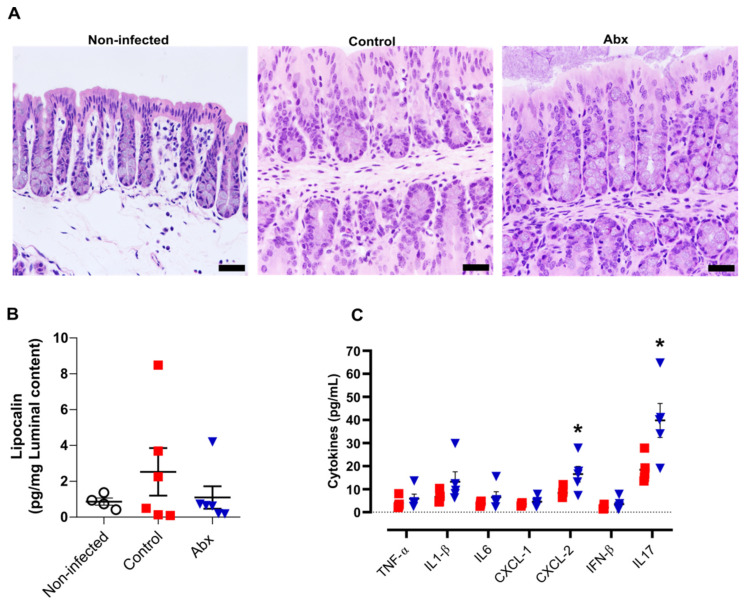
Colon histological and inflammatory alterations after SARS-CoV-2 infection. (**A**) Representative images of hematoxylin and eosin staining of colon sections from each experimental group. Scale bar = 100 µm. (**B**) Concentration of the inflammatory protein lipocalin-2 in luminal content samples from mice as measured by ELISA. (*n* = 4–6). (**C**) Concentration of cytokines in colon homogenates, as measured by ELISA. * *p* < 0.05,Student’s *t*-test. (*n* = 4–5).

**Table 1 cells-11-02572-t001:** Parameters and point scale to calculate the clinical score ^1^.

Clinical Parameters	Degree	Score Points
**Body weight loss**	Normal	0
	<5%	1
	6–10%	2
	11–15%	3
	16–20%	4
	>20%	5
**Appearance**	No piloerection	0
	Piloerection	1
**Spontaneous behavior**	Alert	0
	Slow-moving	1
	Lethargic	2
	Immobile	3
**Eyes**	Normal	0
	Squinted	1
	Closed	2
**Provoked behavior**	Quickly moves away	0
	Slow to move away	1
	Does not respond	2
**Breathing**	Normal	0
	Elevated	1

^1^ Adapted from Moreau et al. [30] and Kumari et al. [31].

**Table 2 cells-11-02572-t002:** Sequences of primers used in qRT-PCR.

Gene ^1^	Sequences
*Tnfa*	F: 5′-TCT TCT CAT TCC TGC TTG TGG C-3′ R: 5′-CAC TTG GTG GTT TGC TAC GAC G-3′
*Il1b*	F: 5′-GGC AGC TAC CTG TGT CTT TCC C-3′ R: 5′-ATA TGG GTC CGA CAG CAC GAG-3′
*Cxcl2*	F: 5′-GGGACAAATAGCTGCAGTCGG-3′ R: 5′-CTACTCTCCTCGGTGCTTAC-3′
*Cxcl11*	F: 5′-CCGAGTAACGGCTGCGACAAAG-3′ R: 5′-CCTGCATTATGAGGCGAGCTTG-3′
*Ifna*	F: 5-CCTGAGAGAGAAGAAACACAGCC-3 R:5-TCTGCTCTGACCACYTCCCAG-3
*Ifng*	F: 5-ACTGGCAAAAGGATGGTGAC-3 R: 5-TGAGCTCATTGAATGCTTGG-3
*Ifnl2/3*	F: 5-AGC TGC AGG CCT TCA AAA AG-3 R: 5-TGG GAG TGA ATG TGG CTC AG-3
*Oasl1*	F: 5-GGATGCCTGGGAGAGAATCG-3 R: 5-TCGCCTGCTCTTCGAAACTG-3

^1^ (Exxtend, São Paulo, Brazil).

## Data Availability

The raw reads of 16S rRNA sequencing were submitted to the National Center for Biotechnology Information’s Sequence Read Archive (NCBI SRA) with accession number PRJNA858922.

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
