# Peer review of "Impact of Microbiota Depletion by Antibiotics on SARS-CoV-2 Infection of K18-hACE2 Mice"

_cells, 2022, doi:10.3390/cells11162572_

Round 1
Reviewer 1 Report
The manuscript is brief and clear. Some minor details in grammar would improve presentation.
Conclusions are clear regarding the role of microbiota in this murine model.
The number of authors is notable.
Author Response
Thanks for reviewing our study. We revised the text as suggested.
Reviewer 2 Report
I don't have further comments.
Author Response
Thanks for reviewing our study.
Reviewer 3 Report
In this research study, Rodrigues et al., evaluate the effect of oral antibiotic mediated depletion of gut bacteria on SARS CoV-2 pathogenesis in mice. After short antibiotic regimen, the authors observed that the pathogenesis of CoV-2 remains unchanged in terms of survival, weight loss or viral burden compared to control mice. At lower dose, authors observed slightly better clinical score in antibiotic treated group, but no changes were noted in lung pathology. There were differences in few cytokines such as IL-1b and IFN-a in the lungs and T cells in the BALF, however overall, the lung milieu appeared similar between the two groups. Likewise, no major changes were observed in the gastrointestinal tract except for an increase in IL-17 and CXCL-2.
Overall, this is a nicely done study with appropriate controls and authors look at appropriate parameters. The authors not only examine the lungs, the obvious target tissue but also the gut tissue which is proximal to the gut bacteria. The results section and discussion section are nicely written with appropriate references wherever necessary. However, the findings of this study appear underwhelming as no remarkable changes are observed in viral pathogenesis. Since, in multiple viral infection models, antibiotic mediated microbiota depletion has been shown to exacerbate the disease, these findings are different and appear not as exciting. Nevertheless, it is important to publish negative or less exciting data for several reasons including to point the scientific community to the right direction for future studies to thoroughly address the role of microbiota in CoV-2 pathogenesis.
I have following comments for authors to consider:
1) In Figure 1E, authors look at viral burden in different tissues. It will be important to look at brain, because in K18 model, lethality has been suggested to occur due to infection of the brain (PMID 33257679). Also, it will be good to add data on nasal wash along with lungs, to determine if replication at early sites is affected.
2) How do authors reconcile the data in Figure 1B and 2A where they observed less clinical score but no gross changes in lung pathology?
3) In Figure 2C, authors note decrease in CD8 and CD4 T cells in the BALF of antibiotic treated mice. It will be important to check if these cells are virus-specific by stimulating them with immunodominant T cell peptides. Since BALF might have less cells, lungs can be used as an alternative. Is reduction in T cell numbers reflected at site of priming like the mediastinal lymph nodes?
4) In Figure 2D, authors look at the cytokine response at day 5 post-infection, which might be slightly late in the pathogenesis. Some of these cytokines peak early at day 2 or 3 post-infection and hence it will be good to look at early time-points (PMID 33257679).
5) Along with cytokines evaluated in this study, interferon lambda has been implicated in CoV-2 pathogenesis and has also been shown to be regulated by changes in microbiota (PMID: 34724828, 35135717). Did authors see any differences in the interferon lambda levels in the lungs between Abx treated and control group?
6) Although K18 mice are widely used in understanding CoV-2V pathogenesis, the obvious limitation might the non-physiological expression of human ACE2 entry receptor. There is likelihood that the detrimental phenotype may be masked by this highly susceptible model that causes 80% lethality in control group. Did authors try using C57BL/6J mice for their study?
7) Another major limitation of this study might be the dose of antibiotic and the magnitude of gut bacterial depletion. Authors have used similar antibiotic regimen in previous studies, however in many other studies, the amount of antibiotic added in drinking water to deplete the gut bacteria is at least 10 times more than that used in this study. The 5-fold decrease in gut bacterial DNA levels is not remarkable, as studies have shown 6 to 8 log differences in 16s RNA levels upon certain antibiotic treatment (Winkler et al; 2020, PMID: 32668198). This is key because, maybe the gut bacteria are reduced only partially by the regimen used in this study. Have authors considered trying antibiotics at higher concentration in drinking water or simply oral gavaging the mice with antibiotic cocktail?
Author Response
Thanks for reviewing our manuscript and for the suggestions.
1) In Figure 1E, authors look at viral burden in different tissues. It will be important to look at brain, because in K18 model, lethality has been suggested to occur due to infection of the brain (PMID 33257679). Also, it will be good to add data on nasal wash along with lungs, to determine if replication at early sites is affected.
Reply: Thanks for the suggestion. We measured viral burden in different areas of the brain, but we did not find any significant difference. We added this information in the new version of the figure 1 (Figure 1E). Unfortunately, we did not collect nasal washes of the mice.
2) How do authors reconcile the data in Figure 1B and 2A where they observed less clinical score but no gross changes in lung pathology?
Reply: In the figure 1B, we present the results of the infection with 5 x 104 PFU/mice in which we analyzed the impact of Abx treatment on infection-associated lethality, while in the figure 2A we checked the lung alterations in mice infected with lower amount of virus (1x104 PFU/animal). Both results indicate that abx-induced dysbiosis do not interfere with SARS-Cov2 infection.
As mentioned by the reviewer, we observed a reduction in the clinical score of the disease on mice treated with Abx. This was associated with an attenuation of some inflammatory parameters including the IL-1b and number of inflammatory cells in the BALF of Abx-treated mice. Despite these effects, no impact on lung pathology was observed. We speculate that the Abx-treatment may reduce translocation of microorganisms from the gut thus impacting on immune cells activation and the clinical score, but this effect may not be sufficient to reduce the SARS-CoV-2 induced damage in the lungs. This aspect is mentioned in the discussion of the manuscript.
3) In Figure 2C, authors note decrease in CD8 and CD4 T cells in the BALF of antibiotic treated mice. It will be important to check if these cells are virus-specific by stimulating them with immunodominant T cell peptides. Since BALF might have less cells, lungs can be used as an alternative. Is reduction in T cell numbers reflected at site of priming like the mediastinal lymph nodes?
Reply: The experiment mentioned is interesting but it would require much more time to be performed.
4) In Figure 2D, authors look at the cytokine response at day 5 post-infection, which might be slightly late in the pathogenesis. Some of these cytokines peak early at day 2 or 3 post-infection and hence it will be good to look at early time-points (PMID 33257679).
Reply: At 5-days post infection we are still in the initial phase of the immune response. We agree that it would be interesting to check an earlier time point but we did not collect samples before 5-days.
5) Along with cytokines evaluated in this study, interferon lambda has been implicated in CoV-2 pathogenesis and has also been shown to be regulated by changes in microbiota (PMID: 34724828, 35135717). Did authors see any differences in the interferon lambda levels in the lungs between Abx treated and control group?
Reply: We measured interferon lambda in lung samples of infected mice but we did not find any difference. We added this information in the revised version of the manuscript (Figure 2E).
6) Although K18 mice are widely used in understanding CoV-2V pathogenesis, the obvious limitation might the non-physiological expression of human ACE2 entry receptor. There is likelihood that the detrimental phenotype may be masked by this highly susceptible model that causes 80% lethality in control group. Did authors try using C57BL/6J mice for their study?
Reply: We did some tests using female C57BL6J and Balb/c mice. However, they were resistant to infection by the strain used in our study.
7) Another major limitation of this study might be the dose of antibiotic and the magnitude of gut bacterial depletion. Authors have used similar antibiotic regimen in previous studies, however in many other studies, the amount of antibiotic added in drinking water to deplete the gut bacteria is at least 10 times more than that used in this study. The 5-fold decrease in gut bacterial DNA levels is not remarkable, as studies have shown 6 to 8 log differences in 16s RNA levels upon certain antibiotic treatment (Winkler et al; 2020, PMID: 32668198). This is key because, maybe the gut bacteria are reduced only partially by the regimen used in this study. Have authors considered trying antibiotics at higher concentration in drinking water or simply oral gavaging the mice with antibiotic cocktail?
Reply: We agree that the dose and the types of antibiotics that we used in our experiments are an important limitation of the study. As mentioned by the reviewer, we used the same antibiotic treatment of our previous study because we observed that with this regimen we had relevant effects on the virus-immune response. Following the reviewer’s comment, we added the following statement in the discussion of the manuscript “Our study has limitations including the absence of analyses on the effect of Abx-treatment during the course of infection, or treatment with more potent Abx-treatment regimens or antibiotics that have been commonly used in COVID-19 patients, such as azithromycin”.
This manuscript is a resubmission of an earlier submission. The following is a list of the peer review reports and author responses from that submission.
Round 1
Reviewer 2 Report
Brito Rodrigues et al. Presented an interesting study about the effect of colon microbiota Antibiotics-related depletion of SARS-CoV-2 infection in K18-hACE2 mice. After treating mice with broad spectrum antibiotics (3 days of treatment), they later intranasal infected mice with SARS-CoV-2 virus. Five days after infection, they analyze lung and colon tissue histology and pro-inflammatory cytokines expression, thus observing that microbiome depletion did not affect SARS-CoV-2 infection in these mice.
Specific comments:
- TITLE:
- Although it is the main result, the title is not attractive, and deviate from the intention to read the manuscript, since the title already explains that there are any effects of SARS-CoV-2 infection.
- INTRODUCTION:
- The introduction is an extremely important paragraph since it supplies of all the necessary basis to understand the results and summarizes the main previously published results to introduce the hypothesis and aim of the presented study. Above more detailed comments:
- In my opinion the authors poorly explained what had been previously reported about gut microbiome and viral respiratory infection, and mainly SARS-CoV-2 infection. This is extremely essential to understand why they are studying the effect of antibiotics-related microbiome depletion in SARS-CoV-2 replication.
- As commented above, they should explain with more details the publication of Zuo et al Gastroenterology 2020.
- Please, cite Reinold et al. about the alterations of microbiome in covid19 patients - Frontiers in Cellular and Infection microbiology- 10.3389/fcimb.2021.747816
- MATERIALS AND METHODS: Antibiotics treatment
- Please add “The” to Addition in the penultimate sentence.
3.1 Microbiota depletion does not change mortality of SARS-CoV-2 infection, but alters clinical symptoms.
- Why did the authors used two different viral titers for the mortality and the samples collection tests, especially considering that they did not observe mortality at 5dpi with the 5-fold higher titer? Please explain.
- Did the authors consider the possibility to use even less virus and prolonged the infection during more days, or use higher viral titer to exacerbate the results?
- Did the authors evaluate the microbiota status at the timepoint of samples collection (5dpi)? If not, how can they ensure that the effects they observed are really related to microbiome depletion. It could be extremely helpful to better understand the results.
- Have the authors an idea of what determine this higher deterioration in no-treated mice related to Abx-treated? Is quite curious since I will aspect a lower clinical score in Abx-treated mice than the untreated one.
- Figure 1D. In this plot they evaluated the clinical score of untreated or Abx-treated mice after SARS-Cov-2 infection. How they calculated the clinical score? Is it an average of the clinical score of each clinical parameter and observation they considered? Please specify. Does this score also include the body weight loss?
3.3. Antibiotic treatment increases inflammatory cytokines in the colon but this effect is not associated with significant alterations in colon histopathology.
- If I have well understood, for their protocols of mice infection, the authors used, as reference, the paper of W. Dong (reference 36). Dong et al used the same animal model however they reached an extremely high level of infection, with mice’s death in around 5-6 days by infecting with 2x104 In Brito Rodriguez’s paper, around 20% of the mice could survive up to 12dpi when infected at 5X104 PFU, thus suggesting a low level of infection. This could cause a limited dissemination of the infection itself, especially in colon (differently from the ref 36), limitedly the effect on colon histopathologic. Considering that they finally decided to infect with 5-fold less virus, they should reach more mice survival after 5dpi. Did the authors evaluate the rate of survival when infected with less viral titer? Did the author try to analyze tissue and cytokine expression at later timepoint? Considering that the infection did not seem extremely effective, as such as reported in ref 36, maybe the effects could be more evident at later timepoints due to prolonged virus replication.
- Higher titer of Neutrophils chemoattractant CXCL2, CXCL1, as well as IL-1β have been already described in the colon of mice with antibiotic-induced dysbiosis. Moreover, they observed a decrease of the surface expression of the CXCR2 receptor on the neutrophils from antibiotic pre-treated mice, thus potentially explaining their inability to migrate to the site of infection (watanabe et al. Plos Pathogen- 10.1371/journal.ppat.1006513). Please comment this paper.
- DISCUSSION
- Although the results demonstrated that microbiome depletion by broad spectrum antibiotics did not affect SARS-CoV-2 infection, in my opinion, it's better to start a paragraph with positive sentences rather than negative one. I’d introduce the paragraph first with the already published (with Pseudomonas, influenza and RSV) and later introduce the study's results.
- The authors affirm that the better clinical score and the reduction of IL1-b expression could be indicative of a “protective effect” of the colon microbiota depletion towards SARS-CoV-2 infection. This affirmation is quite ambitious, since these two observations could be not sufficient to justify a protective effect. Of note, they did not observe histological differences between infected mice with normal or depleted microbiome.
- How can the authors explain this relative higher viral RNA titer in lumen with similar viral RNA in colon cells? Being the virus intranasally administrated, could the virus accidentally pass into the gastrointestinal tract at administration? Moreover, the authors very poorly commented the lumen result in the specific paragraph. They should reinforce it if they want to later discuss it.
- Please, add the study limitations.
Reviewer 3 Report
The content is interesting. The scope is limited as the antibiotics' effects are measured just in an acute administration.
The manuscript should be improved. The relevance in translational medicine is poorly described in introduction and discussion sections. In contrast, conclusions should be clearly sentenced as in a murine model of infection (as is in the title).
The survival of mice should be discussed regarding previous studies in similar conditions.
The minor changes in immune response should be discussed specifically in the sense of they (cytokines in lung and colon) have been reported as modulators of response to SARS-CoV-2.
At the end of discussion limitations (poorly presented) and future proposal to extend scope of this study are desirable. Among these, it could be suggested the use of antibiotics suggested in the early pandemics improving the clinical manifestations. Also, authors could add sentences regarding the putative consequences of their observations in clinical conditions.
Round 2
Reviewer 1 Report
The manuscript has been improved. I understand some of the experiments (e.g. the BALF and tissue/blood leukocyte counts)may not be feasible at this point due to limit access to new samples. It would be worthwhile to include those experiments in your future studies. I still think it would be critical to figure out the "leaky gut" vs. microbiota effects before making conclusions favoring microbiota. Hope these can be addressed in the future.
Reviewer 2 Report
I would like to thanks the authors for all the clarifications and all the changes and additions they made in this revised and improved version of their manuscript.